# CRISPR/Cpf1-mediated DNA-free plant genome editing

Hyeran Kim[1], Sang-Tae Kim[1], Jahee Ryu[1], Beum-Chang Kang[1], Jin-Soo Kim[1,2] & Sang-Gyu Kim[1]

Cpf1, a type V CRISPR effector, recognizes a thymidine-rich protospacer-adjacent motif and induces cohesive double-stranded breaks at the target site guided by a single CRISPR RNA (crRNA). Here we show that Cpf1 can be used as a tool for DNA-free editing of plant genomes. We describe the delivery of recombinant Cpf1 proteins with *in vitro* transcribed or chemically synthesized target-specific crRNAs into protoplasts isolated from soybean and wild tobacco. Designed crRNAs are unique and do not have similar sequences (≤3 mismatches) in the entire soybean reference genome. Targeted deep sequencing analyses show that mutations are successfully induced in *FAD2* paralogues in soybean and *AOC* in wild tobacco. Unlike SpCas9, Cpf1 mainly induces various nucleotide deletions at target sites. No significant mutations are detected at potential off-target sites in the soybean genome. These results demonstrate that Cpf1–crRNA complex is an effective DNA-free genome-editing tool for plant genome editing.

[1] Center for Genome Engineering, Institute for Basic Science, 70, Yuseong-daero 1689-gil, Yuseong-gu, Daejeon 34047, South Korea. [2] Department of Chemistry, Seoul National University, Seoul 08826, South Korea. Correspondence and requests for materials should be addressed to J.-S.K. (email: jskim01@snu.ac.kr) or to S.-G.K. (email: sgkim@ibs.re.kr).

Clustered regularly interspaced short palindromic repeats (CRISPR)–CRISPR-associated proteins (Cas), an adaptive immune system of prokaryotes[1], has now become a powerful tool for genome editing[2–5]. In the type II CRISPR-Cas system, RNase III and the single, large Cas9 protein are involved in the processing of precursor CRISPR RNA (crRNA) in the presence of *trans*-acting crRNA[6]. The Cas9 protein has two additional functions: recognizing the target site and making a site-specific double-stranded break[7]. In the type I and type III systems, several Cas proteins are involved in the recognition and cleavage of target sites[8]. Because of the simplicity and efficiency of the type II system, Cas9 proteins (especially from *Streptococcus pyrogene*) are widely used for genome editing.

CRISPR-Cpf1 (CRISPR from *Prevoltella* and *Francisella*1) has recently been reported as a new type of genome-editing tool[9]; similar to the type II CRISPR-Cas system, a single Cpf1 protein functions in crRNA processing[10], target-site recognition and DNA cleavage[9]. Cpf1, however, differs from Cas9 as follows[9,11]: (1) Cpf1 recognizes T-rich (such as 5′-TTTN-3′) PAM sequences; (2) the PAM sequence is located at the 5′-end of a target DNA sequence, upstream of a protospacer sequence; (3) Cpf1 is guided by a single crRNA, no *trans*-acting crRNA is needed[9]; and (4) Cpf1 is a ribonuclease, processing precursor crRNAs[10]. Among several proteins in the Cpf1 family, LbCpf1 from *Lachnospiraceae bacterium* ND 2006 and AsCpf1 from *Acidaminococcus* sp. BV3L6 act more effectively in human cells compared with other orthologues[9,12].

Previously, we reported a DNA-free genome-editing method in plants using SpCas9 mixed with a single guide RNA (ribonucleoprotein, RNP)[13]. Use of RNPs can reduce off-target effects and cytotoxicity associated with DNA transfection and also avoid the possibility of integration of small DNA fragments derived from plasmids. To test whether the Cpf1 protein can be used as an alternative DNA-free genome-editing tool in plants, we delivered the recombinant LbCpf1 and AsCpf1 proteins mixed with crRNAs into protoplasts isolated from soybean and wild tobacco plants and analysed insertion and deletion (indel) frequencies and patterns at the targeted loci (Fig. 1). The results show that Cpf1–crRNA complexes can introduce targeted mutations in plant genomes.

## Results

**Cpf1–RNP delivery in protoplasts.** We designed nine crRNAs to simultaneously target two homologous genes, *FATTY ACID DESATURASE 2-1A* (*FAD2-1A*, Glyma10g42470) and *FAD2-1B* (Glyma20g24530), in the soybean genome. In our previous Cpf1 study[12], we showed that Cpf1–crRNA complexes could induce mutations at one- or two-base mismatches sites. To avoid off-target effect, we selected crRNAs without allowing three nucleotide mismatches based on the entire homology search in the current soybean reference genome, except the target sites using Cas-Designer (http://rgenome.net)[14] (Fig. 2a and Supplementary Table 1). FAD2 proteins convert oleic acid, a monounsaturated fatty acid, to linoleic acid, a polyunsaturated fatty acid, in seeds[15]. Thus, FAD2 mutations can increase the oleic acid level in soybean oil, a highly desired nutritional trait[16]. We first performed an *in vitro* cleavage assay to examine the activity of Cpf1–RNP complexes, which comprise *in vitro* transcribed crRNAs and recombinant Cpf1 proteins. LbCpf1/AsCpf1–RNPs cleaved the target DNA efficiently *in vitro* (Fig. 2b and Supplementary Fig. 1a).

To monitor the location of Cpf1 proteins in soybean protoplasts, we conjugated a Cy3 fluorophore probe[17] to LbCpf1/AsCpf1 proteins tagged with a nuclear localization signal peptide. Cy3-labelled LbCpf1/AsCpf1 proteins were delivered into soybean protoplasts via polyethylene glycol (PEG)-mediated transformation. After a 24 h incubation, transformed protoplasts were fixed on poly-lysine-coated slides and mounted with 4,6-diamidino-2-phenylindole (DAPI), a nuclear marker, to allow observation of protoplast nuclei. Cy3-LbCpf1 and Cy3-AsCpf1 proteins were found to be predominantly located in the nuclei of soybean protoplasts; the proteins were co-localized with DAPI, but some Cy3-LbCpf1/AsCpf1 proteins remained in the cytoplasm (Supplementary Fig. 1b).

**Cpf1–RNP-mediated gene editing in soybean and wild tobacco.** We next delivered LbCpf1 or AsCpf1 mixed with crRNAs into soybean protoplasts at a 1:6 molar ratio (Cpf1:crRNA) in the presence of PEG in solution[13]. After delivering the Cpf1–RNP complexes, we isolated genomic DNA and performed targeted deep sequencing to analyse indel frequencies and patterns at

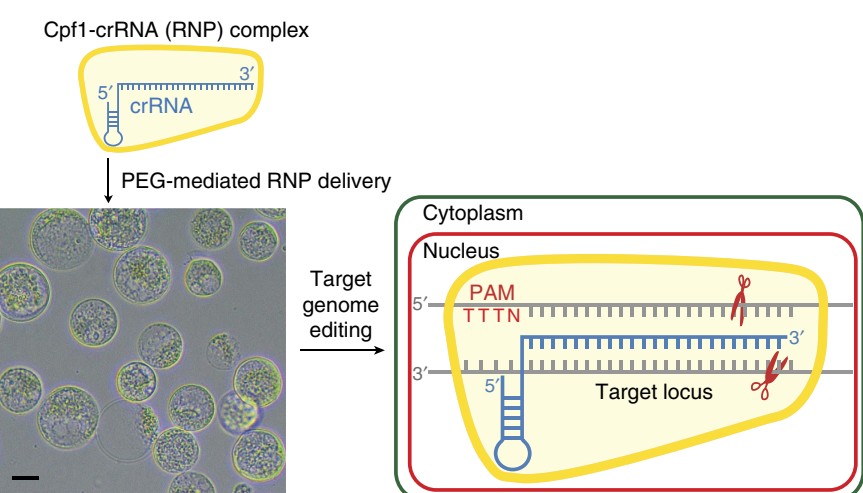

**Figure 1 | Schematic overview of CRISPR/Cpf1–RNP-mediated genome editing in plants.** To edit the plant genome without introducing DNA, recombinant Cpf1 proteins and *in vitro*-transcribed crRNAs were pre-assembled. These active RNP complexes were delivered via conventional PEG-mediated transformation to protoplasts isolated from the target plant. The delivered RNP complex can recognize the crRNA complementary sequence and produce cohesive double-stranded breaks. Scale bar, 10 μm.

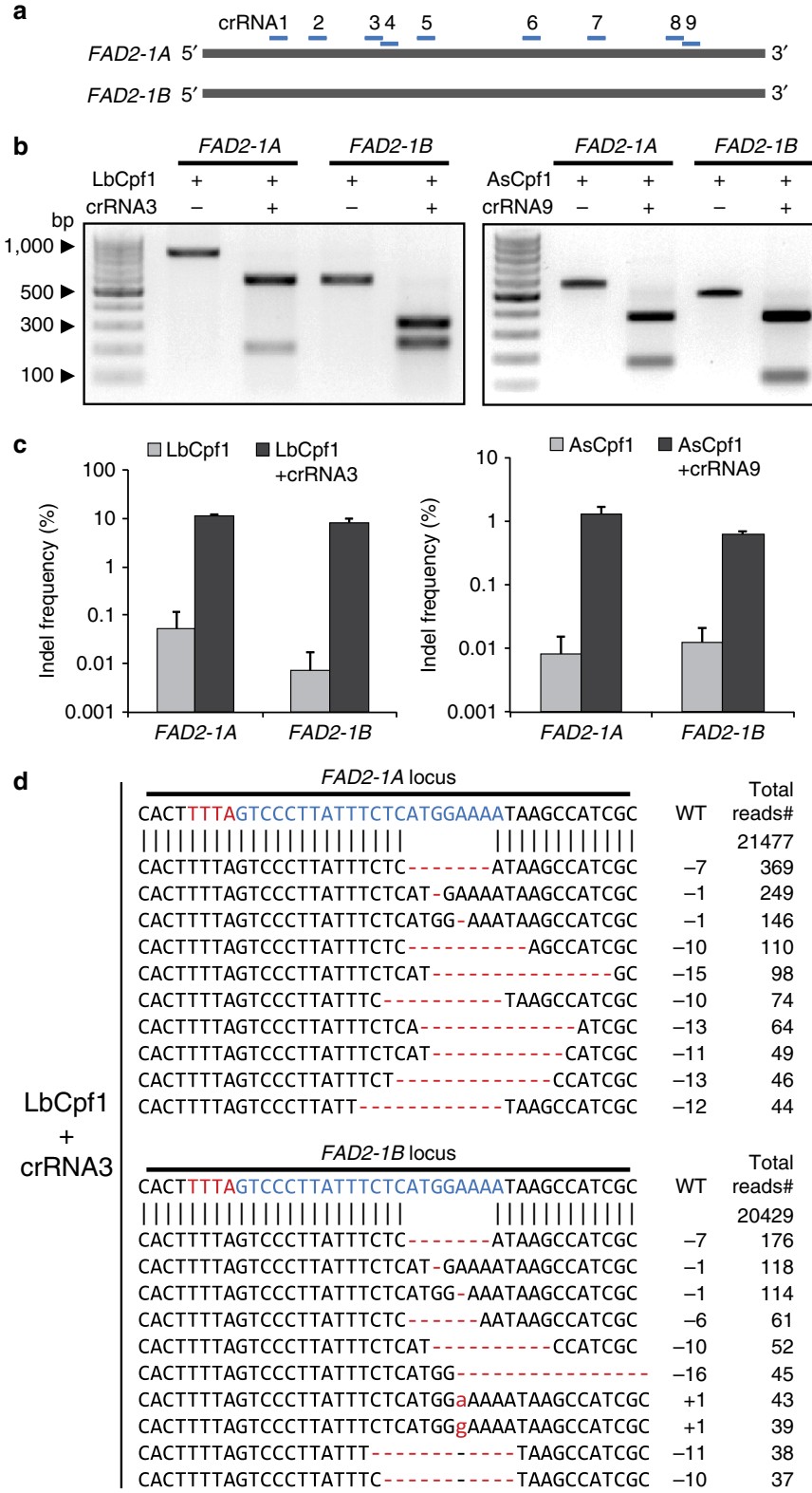

**Figure 2 | CRISPR/Cpf1–RNP-mediated editing of two *GlymaFAD2* genes.** (**a**) The position of nine crRNAs in relation to both *FAD2-1A* and *-1B*. FAD2, FATTY ACID DESATURASE 2. (**b**) The activity of LbCpf1–crRNA3 and AsCpf1–crRNA9 was validated by an *in vitro* cleavage assay. Pre-assembled RNP complexes digested the target amplicons. (**c**) Indel frequencies (%, Log10 scale at *Y* axis) in LbCpf1- and AsCpf1-transformed protoplasts were calculated from targeted deep-sequencing analysis at the two FAD2 target loci. Error bars represent s.d. (*n* = 2). (**d**) Indel patterns at the two target loci in protoplasts treated with LbCpf1–crRNA3. A deletion of seven base pairs was the most common editing pattern at both the *FAD2-1A* and *-1B* loci. Blue, crRNA base-pairing site; Red, PAM sequences.

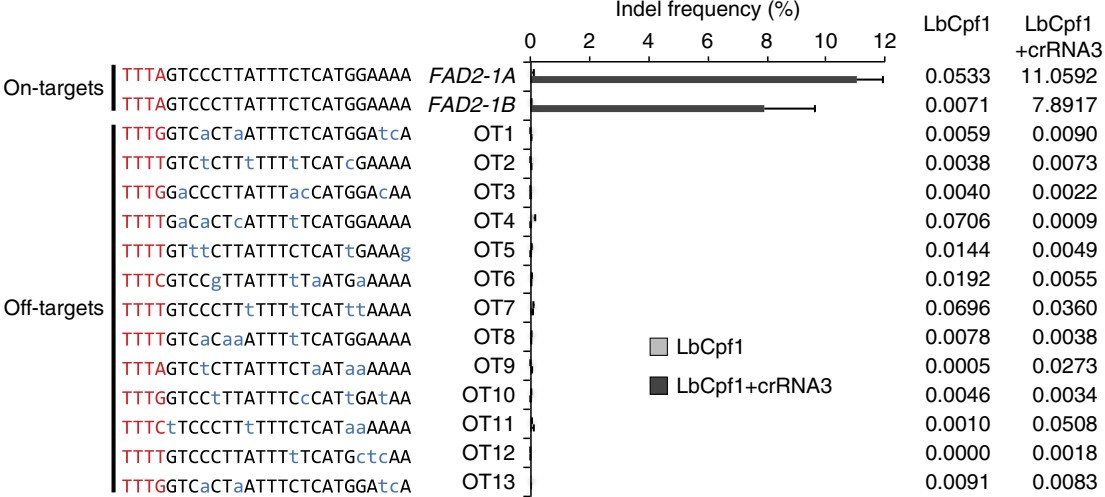

**Figure 3 | In vivo evaluation of LbCpf1–crRNA3 activity at 13 potential off-target sites in the genome.** The indel frequencies (%) at 13 candidate off-target (OT) sites (with up to 4 bp mismatches relative to the crRNA3) were measured and validated in LbCpf1–crRNA3-delivered soybean protoplasts by targeted deep sequencing. No mutations were detected at any of the 13 candidate loci. Error bars represent s.d. ($n = 2$). Blue, mismatched nucleotide bases; Red, PAM sequences.

target sites in the *FAD2-1A* and *FAD2-1B* genes (Fig. 2c,d). Indels were observed at target sites with frequencies that ranged from 0.0 to 11.7% for *FAD2-1A* and to 9.1% for *FAD2-1B* using LbCpf1, and from 0.0 to 1.6% for *FAD2-1A* and to 0.6% for *FAD2-1B* using AsCpf1 in soybean protoplasts (Fig. 2c and Supplementary Fig. 2). Most Cpf1-induced mutation sequences were the result of deletions of several nucleotides (Fig. 2d). We also delivered LbCpf1/AsCpf1–RNPs (Supplementary Table 1) into protoplasts isolated from the leaves of wild tobacco, *Nicotiana attenuata*, to edit the *ALLEN OXIDE CYCLASE* gene, which encodes a key enzyme for jasmonic acid biosynthesis. All Cpf1–RNP complexes completely cleaved their target sites *in vitro* and most of the Cpf1–RNP complexes induced indels at target sites in *N. attenuata* protoplasts (Supplementary Fig. 3).

**In vivo off-target validation.** To validate the specificity of Cpf1–RNP-mediated genome editing, we surveyed the soybean genome *in silico*; using the Cas-OFFinder programme (http://rgenome.net)[18], we first identified potential off-target sites ranging from four to six nucleotide mismatches (Fig. 3 and Supplementary Table 2). We designed specific primer sets (Supplementary Table 3) to amplify the putative off-target loci from genomic DNA isolated from LbCpf1–RNP-transfected protoplasts and performed targeted deep sequencing. No indel mutations were detected at the examined loci (Fig. 3 and Supplementary Fig. 4), suggesting that Cpf1–crRNA does not tolerate four or more mismatches. These data are consistent with recent results in human and mouse cells[12,19]. We observed indels at relatively high frequencies in some control samples (see dOT21 and dOT27 in Supplementary Fig. 4), which are caused by sequencing errors in AT-rich and A- or T-repeat regions[12].

**Chemically synthesized crRNA-mediated gene editing.** When we analysed the indel frequency and patterns induced by Cpf1–RNP complexes, we found that several bases of DNA were inserted into the target sites with low frequencies (0.0028∼0.0233%) (Fig. 4a). These sequences were identical to part of the crRNA sequence, suggesting that the DNA template

for *in vitro* crRNA transcription might be transfected with the Cpf1–RNP complexes into soybean protoplasts and inserted into the target site. Although we treated the reaction mixture with DNase to remove the DNA template after crRNA synthesis, a small amount of intact or fragmented DNA template might still remain in the solution. To eliminate unexpected integration of DNA fragments in transformed protoplasts, we transfected soybean protoplasts with Cpf1 protein and chemically synthesized crRNAs; the crRNA length (∼ 44 bp) for Cpf1 is much shorter than the length of guide RNA (∼100 bp) for SpCas9. We found that chemically synthesized crRNAs successfully induced indels at target sites with activity similar to that of transcribed crRNAs and eliminated the short insertions (Fig. 4b,c).

**Discussion**

We showed here that Cpf1–crRNA RNP complexes successfully induced indel mutations, mainly deletions of several base pairs, at two targeted loci simultaneously in the soybean genome. To implement Cpf1 as a plasmid-based genome-editing tool for plants, one should consider the host-plant-specific codon usage and choose appropriate promoters to express Cpf1 and crRNA in cells as shown in a recent report[20], but these concerns can be circumvented by using the Cpf1–RNP system. In addition, Cpf1–RNPs can considerably reduce off-target mutations[12]. We cloned plant-codon optimized Cpf1 and mature crRNA into a plasmid that we used to express SpCas9 and guide RNA in protoplasts[13]. However, we failed to induce indels in protoplasts using this system (Supplementary Fig. 5a). Xu *et al.*[20] recently showed the same result in rice; the delivery of a plasmid expressing Cpf1 proteins and mature crRNAs into the cells was not able to induce the targeted mutation. To solve this problem, Xu *et al.*[20] delivered two different types of precursor crRNA and were able to edit the target sites with high levels of indel frequencies. In addition, we found that expression of the Cpf1 protein in protoplasts was not detectable (Supplementary Fig. 5b), suggesting that codon optimization is not the only issue to consider for optimizing Cpf1 protein expression in plant cells.

The Cpf1–RNP system has at least three potential benefits for plant genome editing compared with the Cas9 RNP system.

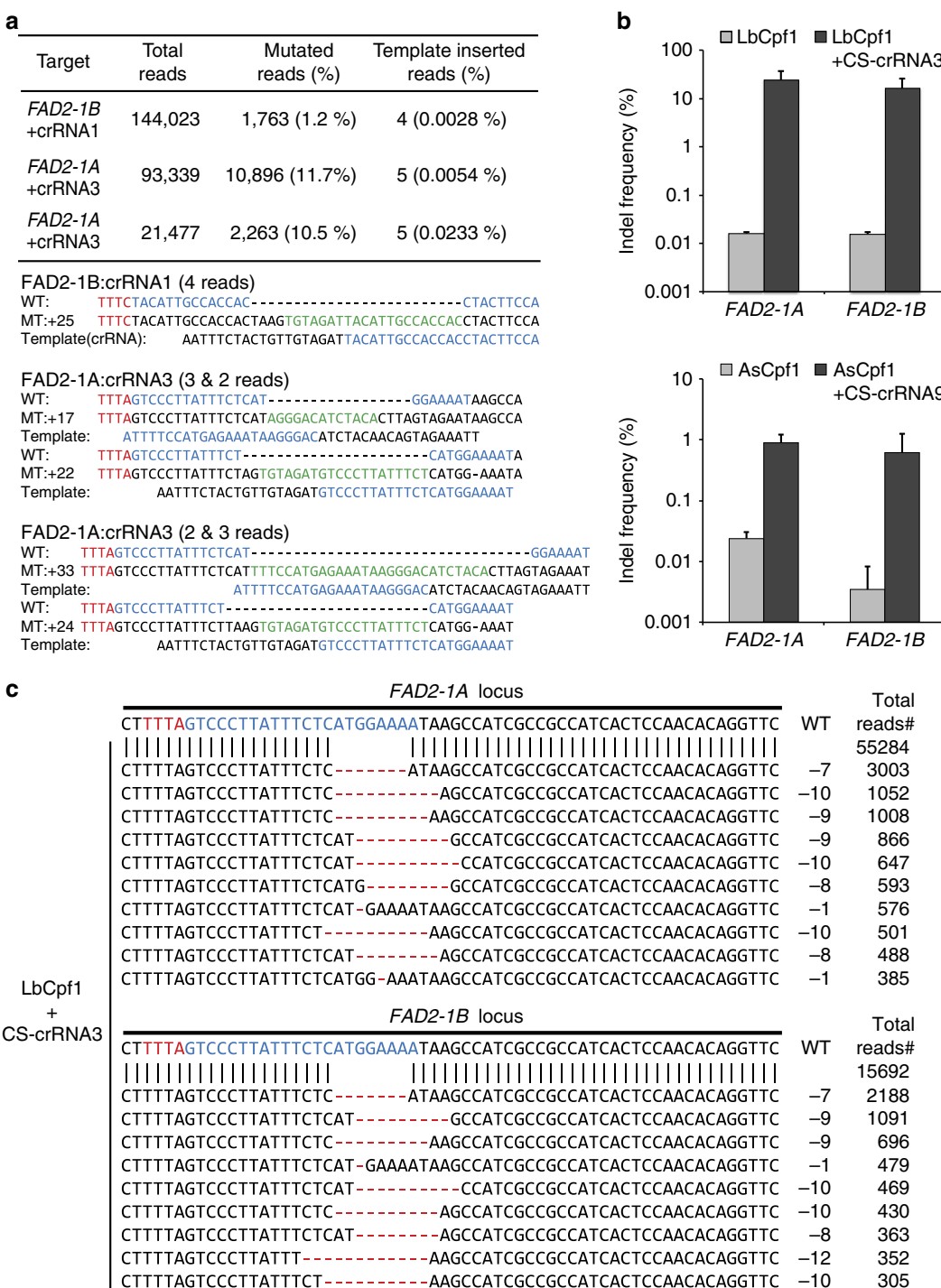

**Figure 4 | Chemically synthesized crRNA-mediated genome editing.** (**a**) Insertion of the template DNA used for crRNA *in vitro* transcription at the target site. (**b**) Indel frequency (%, Log10 scale at *Y* axis) and (**c**) indel patterns at the target sites after delivery of Cpf1 protein and chemically synthesized crRNA3 (CS-crRNA3) into soybean protoplasts. Error bars represent s.d. (*n* = 2). blue, crRNA base-pairing site; green, inserted sequence derived from crRNA template; MT, mutated sequence; red, PAM sequence; WT, wild-type sequence.

First, Cpf1 crRNAs are shorter than Cas9 sgRNAs by ~60 nucleotides, allowing us to use a chemically synthesized crRNA: no foreign DNA was inserted in the host genome using the RNP method when chemically synthesized crRNAs were used. Second, Cpf1 induces larger deletions in the target sites than does SpCas9. Lastly, the cleavage pattern of Cpf1 might assist the NHEJ-mediated insertion of donor DNAs. In our

previous study, we edited a target gene in lettuce protoplasts using the SpCas9-RNP system and successfully regenerated whole plants from the protoplasts. To fully validate the viability of our Cpf1-mediated plant genome editing protocol for producing transgenic plants, we hope to generate whole plants from Cpf1–RNP-transfected protoplasts in the near future and confirm the heritability of mutations. The Cpf1–RNP system will be used

as an additional tool to edit the plant genome without introducing foreign DNA.

## Methods

**Protoplast isolation and PEG-mediated transformation.** *Glycine max* var. William 82 seeds were sterilized and germinated on Murashige and Skoog medium under 16 h light and 8 h dark conditions at 25 °C ± 1 °C in a growth chamber (Koencon, Hanam, South Korea). Seedlings were transferred to 3 L pots 2 weeks after germination. Light was provided by 32 W Osram lamps (170 mol m$^{-2}$ s$^{-1}$). We isolated protoplasts from immature *Glycine max* var. William 82 beans by incubating them with 3xVCP enzymes[21] for 12 h at room temperature. Seeds of wild tobacco, *N. attenuata*, were provided by the Department of Molecular Ecology at the Max Planck Institute for Chemical Ecology in Germany. *N. attenuata* seeds were germinated on Gamborg B5 medium (Duchefa, Biochemie, Harriem, The Netherlands) and 7-day-old young leaves were used for protoplast isolation[13]. PEG-mediated RNP delivery was performed as previously described[13,22]. Briefly, $2 \times 10^5$ protoplasts were mixed with pre-assembled Cpf1/crRNA (1:6 molar ratio) in 300 μl of MMg (4 mM MES, 0.4 M mannitol and 15 mM MgCl$_2$) via an equal volume of freshly prepared PEG solution (40% [w/v] PEG 4000, 0.2 M mannitol and 0.1 M CaCl$_2$). Transfected protoplasts were incubated at 22 °C for 24 h.

**Preparation of recombinant Cpf1 proteins and crRNAs.** His-MBP-tagged Cpf1 proteins (LbCpf1 and AsCpf1) were expressed in *Escherichia coli* and purified by using the Ni-NTA affinity purification method[9]. Briefly, Rosetta cells harbouring Cpf1 plasmids were cultured at 37 °C until OD$_{600}$ = 0.4 and incubated at 18 °C until OD$_{600}$ = 0.6, then induced with 1 mM isopropyl-β-D-thiogalactoside overnight. The cell were harvested and lysed by sonication in 50 ml of lysis buffer (50 mM, HEPES pH 7.0, 200 mM NaCl, 5 mM MgCl$_2$, 1 mM dithiothreitol and 20 mM imidazole) supplemented with lysozyme (1 mg ml$^{-1}$) and protease inhibitor (Roche complete, EDTA-free). The cell lysate was cleared by centrifugation at 13,000 r.p.m. for 30 min, followed by passage through a syringe filter (0.45 μm). The cleared lysate was applied to a nickel column (Ni-NTA agarose, Qiagen), washed with 2 M salt and 20 mM imidazole, and eluted with 250 mM imidazole contained buffer (50 mM HEPES pH 7.0, 200 mM NaCl and 5 mM MgCl$_2$). To conjugate Cy3 (PA13131, GE Healthcare) fluorophores to Cpf1 protein cysteine residues, the Cy3 probe was applied to freshly prepared Cpf1 protein during the purification process[17]. Briefly, Ni-NTA-bound Cpf1 proteins were washed with buffer A (50 mM HEPES pH 7.0, 2 M NaCl, 5 mM MgCl$_2$ and 10% glycerol) and gently mixed with Cy3 probe in DMSO (Fisher Scientific, 1 mg ml$^{-1}$) at a final 1:1 weight ratio overnight at 4 °C in the dark. The Cy3-labelled Cpf1proteins were washed with 10 volume of buffer A and eluted with 250 mM imidazole-containing buffer. The eluted Cpf1 and Cy3-Cpf1 activity were validated by an *in vitro* cleavage assay.

Candidate crRNAs were designed by Cas-Designer[14], which is available at the CRISPR-RGEN Tools website (http://rgenome.ibs.re.kr) (Supplementary Table 1), and synthesized as previously described[13]. crRNA templates were generated by oligo-extension (Supplementary Table 3) using Phusion High-Fidelity DNA polymerase (Finnzymes, Thermo Scientific, Waltham, MA, USA). crRNAs were transcribed *in vitro* with T7 RNA polymerase (New England Biolabs, Ipswich, MA, USA) according to the manufacturer's protocol. The synthetic crRNAs were purchased (Bioneer, Daejeon, Korea) and used for Cpf1-RNP delivery with the same ratio of Cpf1/CS-crRNA (1:6 molar ratio).

**In vitro cleavage assays and targeted deep sequencing.** Soybean and *N. attenuata* genomic DNA was isolated with the DNeasy Plant Mini Kit (Qiagen) and crRNA target regions were amplified with specific primer sets (Supplementary Table 3). The Cpf1 protein (1 μg) and crRNA (300 ng) were pre-mixed at room temperature for 10 min to assemble RNP complexes, which were then applied to cleave the crRNA target amplicon in a reaction buffer (100 mM NaCl, 50 mM Tris-HCl, 10 mM MgCl$_2$, 100 μg ml$^{-1}$ BSA pH 7.9) at 37 °C for 1 h. RNP-digested amplicons were treated with RNase A (4 μg) at 37 °C for 30 min to degrade crRNAs and purified with a PCR purification kit (GeneAll, Seoul, Korea).

After Cpf1–RNP delivery, genomic DNA was isolated from transformed protoplasts. The two target loci were amplified by nested PCR with paralogue-specific primers and subsequently amplified with individual primary primer sets for each crRNA (Supplementary Table 3). Predicted off-target loci were also amplified by specific primer sets (Supplementary Table 3). Multiplexing indices and specific sequencing adaptors were attached to the primary PCR products with PCR using the protocol supplied by the sequencing company (Macrogen, Seoul, South Korea). High-throughput sequencing was performed using Illumina Miseq (v2, 300 cycle) with the paired-end multiplexed library. Raw reads of paired-end Miseq sequencing were joined by the programme 'fastq-join', and indel frequency and patterns in joined reads were analysed using the Cas-Analyzer programme implemented in CRISPR RGEN Tools (http://rgenome.ibs.re.kr).

**Confocal laser scanning microscopy.** The Cy3-conjugated protein was observed with a LSM800 confocal microscope (Carl Zeiss AG, Oberkochen, Germany) equipped with a × 40 objective lens (C-Apochromat × 40/1.1 W). Cy3 and DAPI were excited with 561 and 405 nm laser lines, respectively.

**Cpf1 plasmid construction and expression assay.** The plant-codon-optimized pAsCpf1 and pLbCpf1, and *E. coli*-codon-optimized eAsCpf1 and eLbCpf1 were chemically synthesized (Bioneer, South Korea), and the full sequences of those genes were confirmed by Sanger sequencing. The p2GW7 destination vector was used to transiently express Cpf1 proteins in protoplasts. All plasmid sequences are available in Supplementary Note 1 and accompanied by detailed descriptions.

To assess plasmid-based expression of Cpf1 in plant cells, we applied Cpf1-harbouring plasmids (20 μg) into soybean protoplasts via PEG-mediated transformation. The transformed protoplasts were harvested after 24 h incubation and applied to western blot analysis with anti-HA antibody (sc-7392; Santa Cruz Biotechnology; 1:200) for detecting Cpf1 and anti-Histone-H3 (tri methyl K4) antibody (ab8580; Abcam; 1 μg ml$^{-1}$) for measuring amounts of loading proteins.

**Data availability.** The data supporting the conclusion of this study are available within the article or from the authors upon request.

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

## Acknowledgements

This work was supported by the Institute for Basic Science (IBS-R021-D1). We thank Dr Junho K. Hur for sharing recombinant AsCpf1/LbCpf1 constructs; Dr Seung Hwan

Lee for sharing the protocol for Cy3-protein labeling; Min Kyung Choi and Suji Bae for technical assistance; Dr Daesik Kim for technical discussion; Sejong Choi for providing AsCpf1 proteins; and Drs Suk Weon Kim, Hyun Woo Park and Soon-Chun Jeong for supplying soybean materials. We thank Emily Wheeler, Boston and Heather McDonald for editorial assistance.

## Author contributions

H.K., J.R. and B.-C.K. designed and performed the experiments. H.K. and S.-T.K. analysed deep-sequencing data. H.K., S.-T.K. and S.-G.K. wrote the manuscript. H.K., S.-G.K. and J.-S.K. oversaw the project.

## Additional information

**Competing financial interests:** J.-S.K. and H.K. filed a patent application based on this work. The remaining authors declare no competing financial interests.

