## [Peer Review File · Nature Communications]

Reviewers' comments:

Reviewer #4 (Remarks to the Author):

The authors edited plant genes using Cpf1/gRNA RNP. The advantage of this approach is to avoid introduction of transgenes. This is very important, given that the existence of transgenes makes it very difficult to gain regulatory approval. The group reported successful application of Cas9/gRNA RNP in editing plant genes. Therefore, this paper conceptually is not new. However, the demonstration of the effectiveness of this approach in plants is still valuable and will be of great interests to the plant biology community.

1). I think that the design of the experiments to test off-target effects is fundamentally flawed. All of the potential off-targets have at least four mis-matches. It is expected that editing efficiency would be several magnitude lower than a perfect match. The editing efficiency for perfect match is around 10%. I do not think that the authors' method would be able to detect the low frequency off-target effects. It would be better to test off-target effects with fewer mis-matches.

2) I am confused by the results shown in Figure 2b and SI figure 1. Why in SI figure 1, no distinct digested bands were observed while Figure 2b showed clear patterns? The band sizes do not add up.

Responses to [Reviewer 4]

1) *“I think that the design of the experiments to test off-target effects is fundamentally flawed. All of the potential off-targets have at least four mis-matches. It is expected that editing efficiency would be several magnitude lower than a perfect match. The editing efficiency for perfect match is around 10%. I do not think that the authors’ method would be able to detect the low frequency off-target effects. It would be better to test off-target effects with fewer mis-matches.”*

Please note that there are no such homologous sites with < 4 mismatches in the entire soybean genome and that we chose most highly homologous sites to study Cpf1 off-target effects. We clearly showed that Cpf1-crRNA complex could induce mutations at one- or two-base mismatched sites (especially in the PAM-distal region) in our previous Cpf1 paper (Genome-wide analysis reveals specificities of Cpf1 endonucleases in human cells, Nature Biotechnology, 2016). To avoid this off-target effect, we carefully selected crRNA sequences, which are unique and do not have similar sequences (less than and equal to 3 mismatches) in the entire soybean reference genome. That’s why there is no mismatch sites containing < 4 mismatches in the genome. We provided potential off target sites with 4 mismatches in Figure 3. To avoid off-target effect of Cpf1 and Cas9 in this way, we have developed user-friendly, web-based platform named CRISPR RGEN TOOLS (<http://www.rgenome.net/>). We hope that this reviewer understands our design principle.

2) *I am confused by the results shown in Figure 2b and SI figure 1. Why in SI figure 1, no distinct digested bands were observed while Figure 2b showed clear patterns? The band sizes do not add up.*

We thank this reviewer for this comment. We have updated the supplementary figure1.